# Is Advanced Age a Factor That Influences the Clinical Outcome of Single- or Double-Level MIS-TLIF? A Single-Center Study with a Minimum Two-Year Follow-Up on 103 Consecutive Cases

**DOI:** 10.3390/life13061401

**Published:** 2023-06-16

**Authors:** Daniele Bongetta, Camilla de Laurentis, Raffaele Bruno, Alessandro Versace, Elena Virginia Colombo, Carlo Giorgio Giussani, Roberto Assietti

**Affiliations:** 1Neurosurgery Unit, Ospedale Fatebenefratelli e Oftalmico, 20121 Milano, Italy; 2Neurosurgery Department, Fondazione IRCCS San Gerardo dei Tintori, 20900 Monza, Italy; 3School of Medicine and Surgery, Università degli Studi di Milano Bicocca, 20126 Milano, Italy

**Keywords:** minimally invasive, transforaminal lumbar interbody fusion, MIS-TLIF, PROM, patient’s satisfaction

## Abstract

As life expectancy rises, more elderly people undergo spinal fusion surgery to treat lumbar degenerative diseases. The MIS-TLIF technique, which minimizes soft tissue manipulation, is a promising fusion technique for frailer patients. The aim of this study was to investigate if older age is a significant factor in the clinical outcome of single- or double-level MIS-TLIF. A cross-sectional study was conducted on 103 consecutive patients. Data were compared between younger (<65 y.o.) and older (≥65 y.o.) patients. We observed no significant differences between baseline characteristics of the two groups apart from the frequency of disk space treated, with a relative predominance of L3-L4 space treated in the elderly (10% vs. 28%, *p* = 0.01) and L5-S1 space in younger patients (36% vs. 5%, *p* = 0.006). There was no significant difference in complication rate, surgical satisfaction, EQ 5D-5L, or Oswestry Disability Index (ODI) global or specific scores, with the exception of the EQ 5D-5L “mobility” score, where older patients fared worse (1.8 ± 1.1 vs. 2.3 ± 1.4; *p* = 0.05). The minimal invasiveness of the surgical technique, age-related specific outcome expectations, and biomechanical issues are all potential factors influencing the lack of age group differences in outcome scores.

## 1. Introduction

The frequency of musculoskeletal system degenerative diseases is increasing as life expectancy rises [1,2]. Nevertheless, patients desire to age while remaining physically active [3]. Both younger and older people may experience disabling symptoms from degenerative disorders of the lumbar spine, which in certain situations necessitate invasive surgery such as interbody fusion. Numerous fusion techniques have been proposed and researched [4]. Transforaminal lumbar interbody fusion (TLIF) has gained prominence recently after being shown to be successful, mostly in younger populations [5]. When dealing with senior patients undergoing fusion surgery, spine surgeons usually worry about comorbidities, hospitalization resilience, and mobility issues. According to recent literature, some studies have shown that older age increases the risk of mortality and surgical complications, while others have reported that age alone does not predict a poorer outcome [6,7,8,9,10,11,12]. The minimally invasive version of TLIF (MIS-TLIF) appears to provide equally effective lumbar stabilization while reducing muscle dissection and bleeding, potentially making it the procedure of choice for the elderly [13]. The goal of our study was to investigate if older age is a significant factor in the clinical outcome of single- or double-level MIS-TLIF in a consecutive, monocentric series.

## 2. Materials and Methods

### 2.1. Subjects of Study

Patients who underwent primary, elective surgery for single- or double-level MIS-TLIF for degenerative pathology at our facility from January 2018 to December 2019 were included in this retrospective study. Low back pain with or without radiculopathy and overt biomechanical instability (low-grade spondylolisthesis, isthmic lysis, radiographic dynamic instability) were indications for surgery. Patients of any age were included, and they were divided into younger (those under 65) and older (those who are 65 or older) patients. The STROBE reporting guidelines were adhered to [14].

We gathered demographic information (sex, age at surgery), surgical characteristics (single- or double-level, length of surgery, intra- or post-operative complications), and clinical features (general health status and comorbidities evaluated by means of The American Society of Anesthesiologists—ASA—physical status classification system, length of stay) from clinical charts and intraoperative surgical reports [15].

Two authors conducted telephone interviews with patients at least two years after surgery to assess the clinical outcome and the patient’s satisfaction. Every patient included verbally agreed to fill out the questionnaires, and the answers were guaranteed to be anonymous. The Oswestry Disability Index (ODI) and EQ-5D-5L scores were used to assess the patients’ outcomes, and at the conclusion of the interview, it was specifically asked of the patients if they would have undergone surgery again [16,17,18,19].

The ODI questionnaire specifically examines the perceived level of disability in ten daily living activities, including pain intensity, personal care, lifting, walking, sitting, standing, sleeping, sexual activity (if applicable), social interaction, and travel. Each item has six statements and is scored between 0 and 5. Zero denotes the least degree of disability and five the greatest.

The EQ-5D-5L is a self-assessed, health-related, quality of life questionnaire. The scale measures quality of life on a 5-component scale including mobility, self-care, usual activities, pain/discomfort, and anxiety/depression. Each level is given a rating based on a scale that describes the severity of the related problems (for example, “I have no problems moving about”, “Slight problems”, “Moderate problems”, “Severe problems”, or “I am unable to walk”).

Non-degenerative conditions (trauma, neoplasm, or infection) as the initial instability requiring TLIF, lack of information on surgical or clinical conditions, and refusal to participate in the study through responding to the telephone interview were the exclusion criteria.

### 2.2. Surgical Procedure

Under C-arm X-ray guidance, all interventions were carried out by the same team of highly specialized spinal surgeons. The following is a step-by-step description of the technique we employed [19].

After general anesthesia induction, patients are positioned on a radiolucent table. Typically, two off-midline incisions are made; prior to the incision, local anesthetics and adrenaline (2% mepivacaine with 1:100,000 adrenaline) can be injected into both sides. After the skin has been incised, without using monopolar cautery, a sharp fascia incision is made in order to prevent severe scarring and/or devascularization. The muscle planes are separated longitudinally using blunt dissection with a finger to create a surgical corridor aimed at the transverse apophyses. Ideally, the avascular plane between the multifidus and longissimus dorsi should be followed during subfascial dissection (Figure 1). At the intersection of the transverse process and superior facet, Jamshidi needles are then introduced through these muscular corridors. A guidewire is introduced through the Jamshidi needle after the trocar has been removed. We usually insert screws prior to bone decompression in order to use screw-based dedicated distraction devices, particularly in advanced degenerative stages. Tubular or latero-medial dilators are utilized depending on the hardware set chosen and/or the surgeons’ preferences. For the additional bony steps, microscopic vision is always used. Depending on the surgeon’s preference and the patient’s habit, the surgical bed may be tilted up to 30° away from the operator in order to improve surgical field vision. A nearly full facetectomy is performed using drill and Kerrison punches. To improve thecal sac decompression, a partial laminectomy extension might be added if it is clinically necessary, but the medial insertion of the multifidus is never violated. The disk space and the thecal sac should be visible on the lower and medial sides, respectively, at the conclusion of bone decompression. The exiting nerve root will lie in the upper half of this space, also known as Kambin’s triangle. Curettes, pituitary forceps, and shavers are used in combination to carry out a complete discectomy. Before implanting the cage, we do not usually pack a bone graft in the anterior disk space. Furthermore, we never use any non-autologous osteoinductive or osteoconductive materials. No “banana”-shaped, extendable, or peek cages were used in this case series. One titanium cage in the shape of a bullet was placed in each treated disk space. Finally, the proper size pre-contoured lordotic rods are inserted. After complete hemostasis has been achieved, the previously cut fascia is sutured with continuous absorbable sutures, being careful to spare the muscle fibers, which are allowed to reapproximate on their own. We never use surgical site drains or place a prophylactic vesical catheter. Early mobilization of the patients was encouraged 12–24 h after surgery, without any lumbar corset.

### 2.3. Statistical Analysis

The data were first analyzed as a whole before being stratified by age groups. All data are presented as the mean ± standard deviation. The differences in the continuous variables between groups were assessed using an unpaired *t*-test. To compare categorical data, Fisher’s exact test was used. All tests were two-tailed, and a *p*-value < 0.05 was set to represent statistical significance. GraphPad Prism version 6.04 for Windows, GraphPad Software, La Jolla, CA, USA, was used for statistical analyses.

## 3. Results

We examined the records of 110 patients: Three patients refused to participate in the study, and four were lost during follow-up, resulting in 103 patients included with a 6.8% attrition rate.

We observed no differences in gender distribution, surgical time, ASA score, or complication rate between groups. There was a trend toward significance in the number of disk spaces treated, with older patients being treated at more than one space (22% younger vs. 39% older patients, *p* = 0.09), and in having a longer length of stay (LOS) (4.6 ± 1.9 younger vs. 5.3 ± 1.8 older, *p* = 0.08).

Table 1 summarizes the baseline and clinical–surgical characteristics.

We found significant differences in the frequency of disk space treated, with a relative predominance of L3-L4 space treated in the elderly (10% younger vs. 28% older patients, *p* = 0.01) and L5-S1 space in younger patients (36% younger vs. 5% older patients, *p* = 0.006) (Figure 2).

More generally, when the disk spaces treated were divided into the upper lumbar region (ULR: L2-L3 and L3-L4 disks) or lower lumbar region (LLR: L4-L5 and L5-S1 disks), we found a significant difference in the percentage of younger patients treated at lower spaces (87% younger vs. 67% older patients, *p* = 0.008) (Figure 3).

Concerning complications, there were no dural tears, surgical site infections, or wound dehiscences. Instead, there were four post-operative hyperpyrexia cases requiring antibiotic administration (3.9%), all in the younger group, and two screw malposition cases requiring revision (1.9%).

Table 2 summarizes follow-up patient-reported outcome data.

The minimum and maximum lengths of follow-up were 24 and 46 months, respectively. The overall surgical satisfaction rate was 78% (76% younger vs. 84% older patients, *p* = 0.4) (Figure 4).

There was no significant difference between groups for FU duration, surgical satisfaction, EQ-5D-5L scores, or ODI global or specific scores, with the exception of the EQ-5D-5L “mobility” score, where older patients fared worse (1.8 ± 1.1 younger vs. 2.3 ± 1.4 older patients; *p* = 0.05) (Figure 5).

## 4. Discussion

It is complicated to evaluate the outcome of spinal surgery. Many variables, ranging from radiological or clinical ones to quality of life (QOL) assessments, could be included [20]. The timing of the evaluation is also a variable, as it may play a role because an early one may suffer from post-operative issues, whereas a long-distance one may be biased by the progressive nature of the degenerative disease [21]. The purpose of this study was to investigate how age affects the clinical outcome of single- or double-level MIS-TLIF. To make things simpler, we devised a cross-sectional study that relied solely on Patient Reported Outcome Measures (PROMs) at a minimum of 2 years and a maximum of 4 years for follow-up, in order to avoid follow-up timing biases. Firstly, we selected multi-item, more complicated scores, such as the ODI and, more specifically, the EQ-5D-5L, whose combination of several health dimensions and severity levels generates a total of 3125 (5^5^) unique health states, in order to gather as much information as possible from the reported results. On the other hand, we also asked a simple question: “Are you satisfied with the results of your surgery?”

With both the more complex and simpler approaches, our findings revealed no significant differences in satisfaction between younger and older patients, with an overall satisfaction rate of 78% (76% younger vs. 84% older patients, *p* = 0.4). As a matter of fact, while multi-item instruments give a more accurate picture of treatment outcome, a single-item global assessment may be an aggregate of all significant dimensions from the patient’s perspective, which may help to explain why more sophisticated PROMs were unable to detect significant differences. One minor exception was in the EQ-5D-5L questionnaire’s “motility” domain, where older patients fared significantly worse (1.81.1 younger vs. 2.31.4 older patients; *p* = 0.05). This could be due to the reported increased tendency to receive multilevel treatment (22% younger vs. 39% older patients, *p* = 0.09) or to the aging process itself (other lower limb joint degenerative diseases, less stamina, poor cardiovascular reserve), as reflected also in the trend for a longer LOS (4.61.9 younger vs. 5.31.8 older, *p*= 0.08).

Surprisingly, this motility impairment, combined with overall senile frailty, did not result in a lower outcome score in senior patients. Several potential explanations for the lack of age group differences in outcome scores might be postulated, such as the surgical technique’s minimal invasiveness, age-related specific outcome expectations, and biomechanical issues.

From a surgical perspective, using a minimally invasive technique may have had some benefits in reducing the potential age-related burden. The small skin incision size and retractors’ pressure strain are the first advantages of MIS-TLIF, but histological, electromyography (EMG), and magnetic resonance maging (MRI) analyses also showed how little damage is caused to the muscles [22,23,24,25]. According to our observations, this in particular has a number of benefits. The systematic preservation of multifidus muscle medial attachment, as well as the sparing of supraspinous and interspinous ligaments and their associated mechanoreceptors, we believe, aided in early mobilization [26]. Furthermore, due to limited soft tissue manipulation, we were able to follow a no-drain, no-corset policy, which may have reduced complication rates and accelerated recovery, particularly in elderly patients. In fact, evidence that minimally invasive approaches are superior to open approaches is constantly reported in the literature, particularly in frail, problematic patients such as obese and elderly patients [27,28]. New visualization systems such as endoscopy might also play a role in this regard [29].

Secondly, post-operative impairment assessment should be interpreted in light of the baseline mobility issues of senior patients. In fact, for a senior patient, some mobility impairment and postoperative limitations, such as those highlighted by the EQ5-5D-5L, may be more justifiable and acceptable than for a younger patient and do not affect overall satisfaction with the surgical procedure. Indeed, literature has shown that patients’ surgical expectations can predict improvements in functional outcome scores after surgery [30]. To avoid patients’ disappointment, honest and thorough preoperative counseling about the potential complications and foreseeable outcome of surgery is then recommended.

Lastly, the significant difference in the relative frequency of lumbar segment treatment could have contributed to the lack of significant differences in PROM scores across age groups. Not surprisingly, the L4-L5 disk was the most frequently treated level in both age groups, as it is known to be more susceptible to axial torsion and the most common site for lumbar instability (52% younger vs. 55% older patients, *p* = 0.9) [31]. On the contrary, the L5-S1 disk space was significantly more frequently treated in younger patients (36% younger vs. 5% older patients, *p* = 0.006), whereas the L3-L4 disk space was more frequently treated in older patients (10% younger vs. 28% older patients, *p* = 0.01). From an anatomical standpoint, the L5-S1 motion segment is deep-set in the pelvis and is covered in iliolumbar ligaments. This prevents torsional strain but makes it more vulnerable to axial compressive forces, which are more typical of younger, more active patients [32]. On the other hand, the more mobile L3-L4 segment is frequently involved in lumbar stenosis (LS) as a result of instability-related ligamentum flavum hypertrophy that is common in elderly patients [33,34]. Indeed, in patients over the age of 65, symptomatic LS is the leading cause of disability and restricted mobility, as well as the most common indication for spinal surgery [33,35]. As a result, we may be comparing two different stages of the lumbar biomechanical impairment spectrum. Senior patients, in particular, may have a longer recovery but may benefit more from the surgical procedure due to the resolution of LS symptoms.

The concept of lumbar spine biomechanical impairment as a spectrum leads to a final thought about aging. In our study, we used the WHO definition of “65 years of age or older” for the elderly. Nonetheless, in the modern era, in which senior citizens are likely more concerned with their well-being and overall physical condition, a clear cut-off for frailer patients based solely on age cannot be established. In fact, in our analyses, we found no difference in the ASA score, as inaccurate as it may be in describing a patient’s fitness. In the end, this study suggests that when it comes to MIS-TLIF short fusion constructs, age is more than just a number, but rather a spectrum of health statuses, implying that careful patient selection is critical for optimizing clinical outcomes.

### Limitations

There are several limitations to this study. First of all, as it is retrospective in nature and has a limited number of cases; additional research that is prospective, more extensive, and ideally multi-centric is required to validate these findings. There are several limitations to this study. First of all, its retrospective nature and small number of cases call for future perspective, more numerous and hopefully multicentric studies to validate these results. Secondly, there are numerous potential intrinsic biases in PROMs that must be considered. Specifically, a recent review identified seven potential bias sources: collection-method-related bias, non-response bias, proxy response bias, recall bias, language bias, fatigue bias, and timing bias [36]. Likewise, we did not include a mental health status analysis of the patients, which could be a source of bias [37]. It is, in fact, well established that positive changes in depression and anxiety are associated with improvements in pain, disability, satisfaction, and overall functioning. Moreover, the surgeons involved were all experienced, with a high-volume case load in spinal fusion surgeries. Hence, the generalization of the results of this study should be considered only after a proper learning curve has been completed. Finally, despite the fact that we only evaluated short fusions, we did not conduct a thorough post-operative imaging study (including sagittal balance analysis and subsidence evaluation).

## 5. Conclusions

The clinical outcome of single- or double-level MIS-TLIF is unaffected by advanced age. The minimal invasiveness of the surgical technique, age-related specific outcome expectations, and biomechanical issues are all potential factors influencing the lack of age group differences in outcome scores. High levels of satisfaction can be reported when the appropriate minimally invasive technique is used in conjunction with careful patient selection and preoperative counseling.

## Figures and Tables

**Figure 1 life-13-01401-f001:**
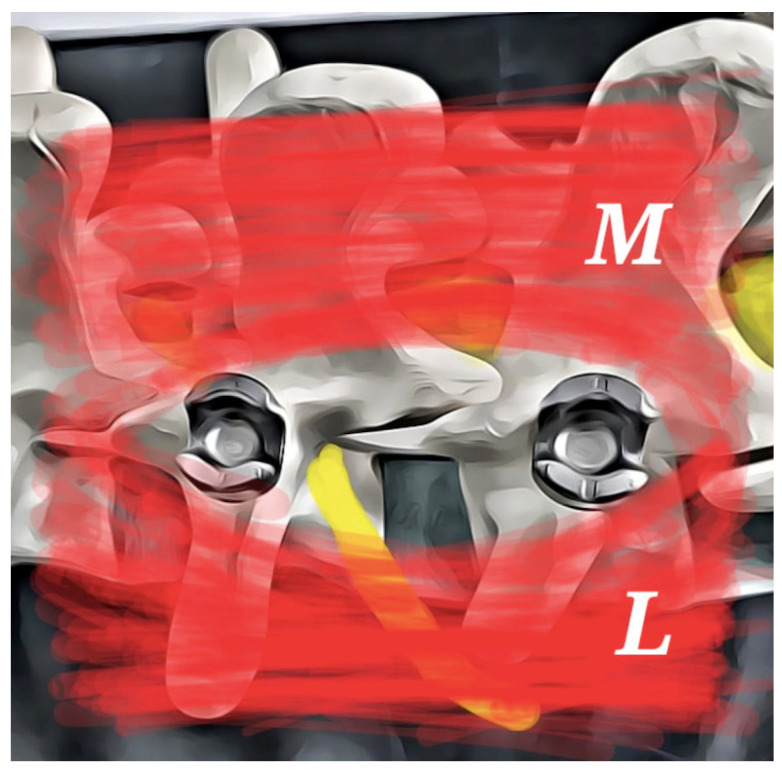
Sketching of a left-side, L4-L5 approach to the Kambin’s triangle via a 30°-tilted, muscle-sparing approach between Multifidus (M) and Longissimus (L) muscles.

**Figure 2 life-13-01401-f002:**
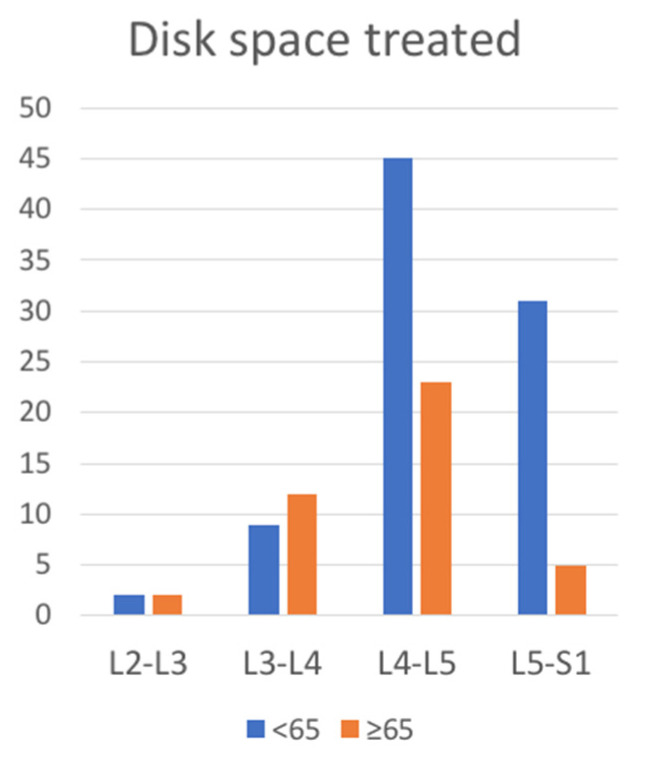
Disk space treated.

**Figure 3 life-13-01401-f003:**
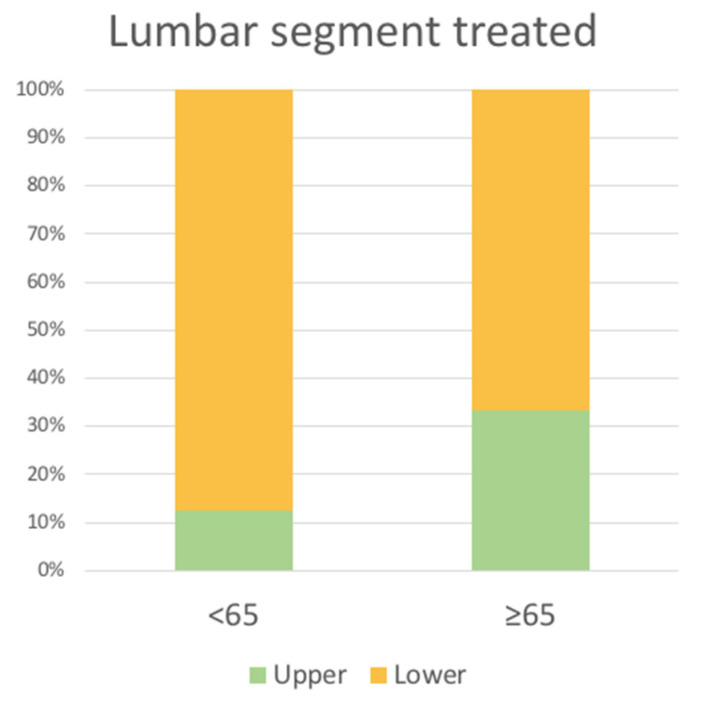
Proportion of lumbar segment treated: upper (L2-L3 and L3-L4) vs. lower (L4-L5 and L5-S1).

**Figure 4 life-13-01401-f004:**
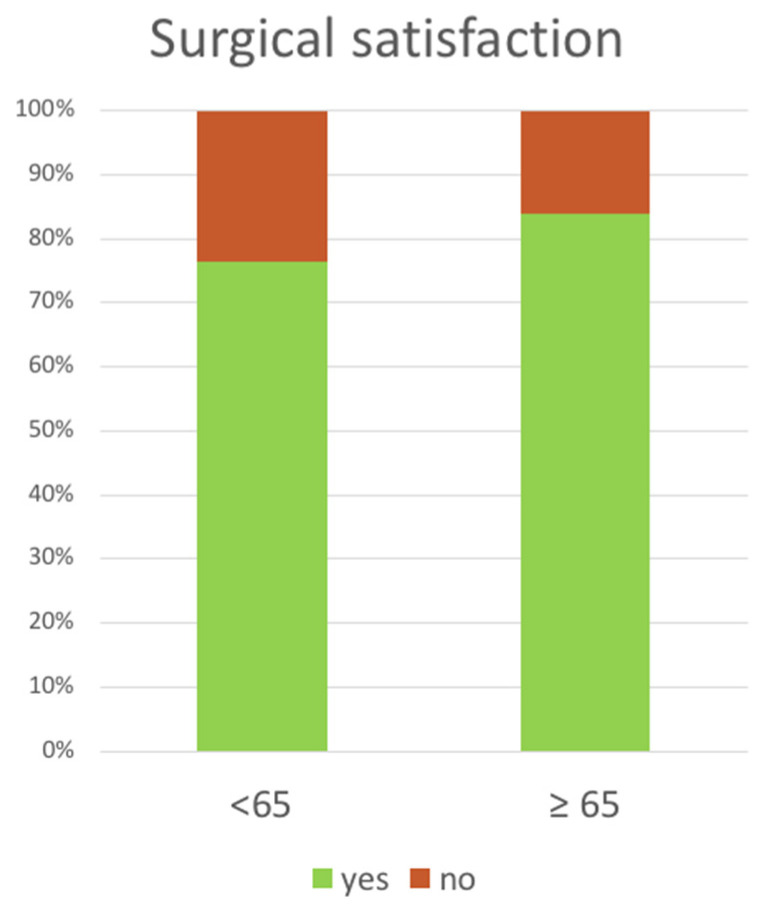
Patients’ self-reported surgical satisfaction.

**Figure 5 life-13-01401-f005:**
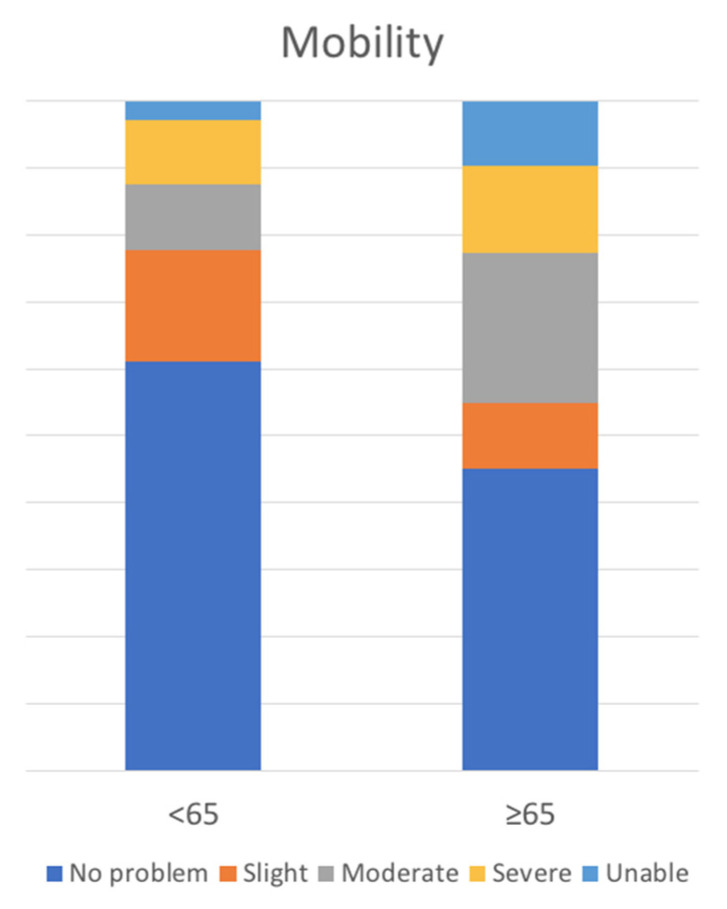
EQ-5D-5L “Mobility” score.

**Table 1 life-13-01401-t001:** Clinico-surgical characteristics.

	Total	<65	≥65	*p*
N	103	72	31	-
m:f	55:48	40:32	15:16	0.5
age (years)	58.5 ± 12.8 (27–84)	52.0 ± 9.1 (27–64)	73.5 ± 5.6 (65–84)	0.0001
ASA score	1.8 ± 1.1 (1–3)	1.7 ± 1.1 (1–3)	2.1 ± 1.2 (1–3)	0.1
one:two levels	75:28	56:16	19:12	0.09
L2-L3	4 (3%)	2 (2%)	2 (5%)	0.6
L3-L4	21 (16%)	9 (10%)	12 (28%)	0.01
L4-L5	68 (53%)	45 (52%)	23 (55%)	0.9
L5-S1	36 (28%)	31 (36%)	5 (12%)	0.006
ULR(L2-L4):LLR(L4-S1)	25:104	11:76	14:28	0.008
surgical time (min)	104 ± 37 (45–220)	103 ± 36 (50–220)	106 ± 41 (45–200)	0.7
LOS (days)	4.8 ± 1.9 (2–13)	4.6 ± 1.9 (2–13)	5.3 ± 1.8 (3–9)	0.08
Complications	4 fever 2 HM	4 fever 1 HM	1 HM	0.7

ULR—upper lumbar region (L2-L3 and L3-L4 disks); LLR—lower lumbar region (L4-L5 and L5-S1 disks); LOS—length of stay (including surgery day); HM—hardware malposition.

**Table 2 life-13-01401-t002:** Follow-up patient-reported outcome measures.

		Total	<65	≥65	*p*
FU duration (months)		34.8 ± 6.1 (24–46)	35.1 ± 6.0 (24–46)	33.9 ± 6.4 (24–45)	
Surgical satisfaction Y:N		80:23	55:17	26:5	
EQ 5D-5L	Mobility	1.9 ± 1.3	1.8 ± 1.1	2.3 ± 1.4	0.05
	Self-care	1.6 ± 0.9	1.6 ± 0.8	1.6 ± 1.0	
	Usual activities	2.2 ± 1.3	2.1 ± 1.2	2.3 ± 1.4	
	Pain/discomfort	2.6 ± 1.3	2.6 ± 1.3	2.5 ± 1.3	
	Anxiety/depression	1.6 ± 1.1	1.6 ± 1.1	1.7 ± 1.2	
	VAS	59.5 ± 23.0	61.1 ± 22.3	55.6 ± 24.7	
ODI	Pain	2.0 ± 1.4	1.9 ± 1.4	2.0 ± 1.5	
	Personal Care	0.7 ± 0.8	0.7 ± 0.8	0.6 ± 01.0	
	Lifting	1.7 ± 1.5	1.7 ± 1.5	1.8 ± 1.6	
	Walking	1.1 ± 1.4	0.9 ± 1.3	1.5 ± 1.7	
	Sitting	1.4 ± 1.3	1.6 ± 1.3	1.1 ± 1.3	
	Standing	1.8 ± 1.5	1.7 ± 1.4	2.2 ± 1.6	
	Sleeping	0.8 ± 1.2	0.8 ± 1.2	0.7 ± 1.3	
	Sex life	0.7 ± 1.2	0.7 ± 1.0	0.9 ± 1.8	
	Social Life	1.3 ± 1.4	1.3 ± 1.3	1.5 ± 1.7	
	Travelling	1.5 ± 1.4	1.3 ± 1.2	1.9 ± 1.7	
	TOTAL%	26.6 ± 20.7	25.4 ± 19.2	29.2 ± 23.8	

FU—follow-up; ODI—Oswestry disability index.

## Data Availability

The data presented in this study are available on request from the corresponding author.

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
