# Peer review of "Is Advanced Age a Factor That Influences the Clinical Outcome of Single- or Double-Level MIS-TLIF? A Single-Center Study with a Minimum Two-Year Follow-Up on 103 Consecutive Cases"

_life, 2023, doi:10.3390/life13061401_

Round 1

Reviewer 1 Report

The authors in this study address a particularly useful and interesting topic. The increase in the average age of the population leads to a proportional increase in degenerative pathologies.

The study, although well structured, presents, in my opinion, some critical issues.

The authors present a fair number of patients affected by degenerative pathology of the lumbo-sacral spine. The individual pathologies are not specified (disc herniation, canal stenosis, listhesis and/or combinations of them) and consequently there is no correlation with the results obtained.

The patients underwent surgical treatment under general anesthesia. Did the authors use frailty rating scales? The presence of comorbidities (cardiac, respiratory diseases, ...) more frequently found in old age could certainly interfere.

Reviewer 2 Report

MIS TLIF has been established as a strategy that has excellent clinical results.  This manuscript is well-written and provides unsurprising conclusions.  The fact that a uniform interbody fusion strategy (one type of cage with autograft) was used sets the authors up for the potential for an excellent radiographic comparison of the 2 groups.  However, this is not present. The clinical outcomes are certainly relevant, but the radiographic outcomes are just as important.  Do older patients fuse as well?  Is the incidence of pseudarthrosis higher in the older group when the same fusion technique was utilized?

Reviewer 3 Report

Authors present a retrospective  study on 103  patients to compare clinical outcome of single- and double level MIS-TLIF, with 65 years of age as a cut off.  L3-L4 space was predominant level treated in the elderly  and L5-S1 space in younger patients. There was no significant difference in complication rate, surgical satisfaction, EQ 5D- 29 5L or Oswestry Disability Index (ODI) global or specific scores, with the exception of the EQ  "mobility" score, where older patients fared worse.

There are several drawbacks of this study - retrospective character, low number of patients, mixed cohort of single and two-level TLIF (not double, I suggest to change this). The clinical outcome cannot be assesed per telephone, and that is also a major problem of this study - I suggest to include "clinical outcome gathered by survey" or "reported clinical outcome"; since real outcome would mean to conduct a proper neurological examination and imaging. Description of surgical technique is too long and belongs to a textbook and not a peer-review manuscript. Were there any differences among one level vs. two level in general and among one level in elderly vs. two level in younger - since these are the most dominant groups? Also, there are no data whatsoever on the indication for surgery - how many patients did have previous spine surgeries, this is crucial. How many cases of first-surgery, degenerative spondylolisthesis? How many surgeons did conduct surgeries? What was the operative time? For Discussion include and comment:

Ma T, Zhou T, Gu Y, Zhang L, Che W, Wang Y. Efficacy and safety of percutaneous transforaminal endoscopic surgery (PTES) compared with MIS-TLIF for surgical treatment of lumbar degenerative disease in elderly patients: A retrospective cohort study. Front Surg. 2023 Apr 17;9:1083953. doi: 10.3389/fsurg.2022.1083953. PMID: 37139262; PMCID: PMC10149668.  

Acceptable. 

Round 2

Reviewer 3 Report

Adequate response to remarks.

Acceptable.

Author Response

Thank you for your reviewing